# Effect of Dried Apple Pomace (DAP) as a Feed Additive on Antioxidant System in the Rumen Fluid

**DOI:** 10.3390/ijms231810475

**Published:** 2022-09-09

**Authors:** Iga Bartel, Magdalena Koszarska, Kamil Wysocki, Martyna Kozłowska, Małgorzata Szumacher-Strabel, Adam Cieślak, Beata Wyrwał, Aleksandra Szejner, Nina Strzałkowska, Jarosław Olav Horbańczuk, Atanas G. Atanasov, Artur Jóźwik

**Affiliations:** 1Institute of Genetics and Animal Biotechnology, Polish Academy of Sciences, 05-552 Jastrzębiec, Poland; 2Department of Animal Nutrition, Poznań University of Life Science, 60-637 Poznań, Poland; 3Department of Pharmacognosy, University of Vienna, 1090 Vienna, Austria

**Keywords:** rumen fluid, dried apple pomace, enzymes, antioxidants, polyphenols, dairy cow

## Abstract

The study aimed to evaluate the effect of dried apple pomace (DAP) as a feed additive on the enzymatic activity and non-enzymatic compounds belonging to the antioxidant system in cattle rumen fluid. The experiment included 4 Polish Holstein–Friesian cannulated dairy cows and lasted 52 days. The control group was fed with the standard diet, while in the experimental group, 6% of the feedstuff was replaced by dried apple pomace. After the feeding period, ruminal fluid was collected. The spectrophotometric technique for the activity of lysosomal enzymes, the content of vitamin C, polyphenols, and the potential to scavenge the free DPPH radical was used. The enzyme immunoassay tests (ELISA) were used to establish the activity of antioxidants enzymes and MDA. Among the rumen aminopeptidases, a significant reduction (*p* < 0.01) from 164.00 to 142.00 was observed for leucyl-aminopeptidase. The activity of glycosidases was decreased for HEX (from 231.00 to 194.00) and β-Glu (from 1294.00 to 1136.00), while a significant statistically increase was noticed for BGRD (from 31.10 to 42.40), α-Glu (from 245.00 to 327.00), and MAN (from 29.70 to 36.70). Furthermore, the activity of catalase and GSH (*p* < 0.01) was inhibited. In turn, the level of vitamin C (from 22.90 to 24.10) and MDA (from 0.36 to 0.45) was statistically higher (*p* < 0.01). The most positive correlations were observed between AlaAP and LeuAP (r = 0.897) in the aminopeptidases group and between β-Gal and MAN (r = 0.880) in the glycosidases group. Furthermore, one of the most significant correlations were perceived between SOD and AlaAP (r = 0.505) and AcP (r = 0.450). The most negative correlation was noticed between α-Gal and DPPH (r = −0.533) based on these observations. Apple pomace as a feed additive has an influence on lysosomal degradation processes and modifies oxidation–reduction potential in the rumen fluid. Polyphenols and other low-weight antioxidant compounds are sufficient to maintain redox balance in the rumen.

## 1. Introduction

The fruit sector uses apples to process into dried apples, baby food, jams, and sauces, but due to the apples’ remnants, the most meaningful is the production of juices (clarified and concentrated), ciders, alcoholic beverages, and vinegar [1,2]. Fruit by-products from apple processing account for 25–30%, and overall, it is known as apple pomace (i.e., peel, core with seed, stem). Hence, it is considered the main industrial waste [3,4]. On the other hand, apple pomace contains valuable nutrients such as carbohydrates (starch, glucose, fructose, sucrose), minerals (P, Mg, Ca, Fe), and dietary fiber. In addition, it is a source of polyphenolic compounds (from 31% to 51%), which protect against diseases related to oxidative stress or inflammation [5]. Both surpluses of apple pomace and nutritional value prompt researchers to seek possibilities using post-industrial waste. One of the approaches is including it as a livestock breed ingredient to enrich and improve the quality of animal food [6]. It has been proven that the inclusion of apple pomace in the ruminant feedstuff enhances the growth and lactation of the animals [7]. Another reason for implementing fodder modification is to increase the efficiency of breeding animals and maintain antioxidative/oxidative homeostasis. Metabolic processes are naturally related to the production of reactive compounds in organisms. However, the disproportion between the production of free radicals and their derivatives and the antioxidant defense leads to the state of oxidative stress. Cows are particularly vulnerable to disturbance of homeostasis and one of the reasons is an intensification of milk yield in recent years [8]. Oxidative stress plays a crucial role in the occurrence of many diseases, reproductive disorders, and reduced milk yield [9,10]. Organisms have antioxidant systems that consist of antioxidant enzymes such as catalase, glutathione peroxidase, and superoxide dismutase, as well as non-enzymatic antioxidants: retinol, tocopherol, ascorbic acid, glutathione, α,α-diphenyl-β-picrylhydrazyl (DPPH), and others [11]. One of the most effective and advanced cell degradation systems is the lysosomal system. Its main role is the elimination of extraneous and endogenous molecules under physiological conditions. Due to the fact that lysosomes are particularly sensitive to oxidative stress, the lysosomal enzymes are one of the most relevant indicators for the assessment of homeostasis between pro-oxidative and antioxidative processes in organisms/cells [12,13,14]. Furthermore, these are useful compounds like malondialdehyde (MDA) as a lipid peroxidation marker [15]. Many studies investigated fermented apple pomace as additives to daily animal food [16,17]. We hypothesized that substances contained in dried apple pomace (DAP) could increase lysosomal activity and enzymatic and non-enzymatic compounds content in the rumen fluid, which would result in reduced incidence of oxidative stress and inflammation in dairy cows. However, little information is available on using dried apple pomace (DAP) as a feed additive and its effect on these parameters. This study aimed to examine whether DAP was affected by ruminal enzymatic (catalase, glutathione reductase, superoxide dismutase) and non-enzymatic (ascorbic acid, glutathione, DPPH) antioxidants content, malondialdehyde (MDA) level, total polyphenols, and lysosomal enzymes activity to verify the hypothesis.

## 2. Results 

The results of the activity of lysosomal enzymes are presented in Table 1. No statistically significant changes were observed for aminopeptidases activity such as alanyl-aminopeptidase (AlaAP) and arginyl-aminopeptidase (ArgAp) except for leucyl-aminopeptidase (*p* ≤ 0.01), where significant inhibition of activity in the experimental group was observed. Statistically, more differences were noticed for enzymes belonging to glycosidases enzymes. The highest statistically significant reduction (*p* < 0.01) from 231.00 to 194.00 was perceived for N-acetyl-hexosaminidase (HEX), while the highest increase (*p* < 0.01) from 31.10 to 42.40 was recorded for β-glucuronidase (BGRD) in the experimental group. Significant (*p* ≤ 0.01), but lower decrease was recorded from 1294.00 to 1136.00 for β-glucosidase (β-Glu) and an increase from 245.00 to 327.00 for α-glucosidase (α-Glu). The lowest but significant increase was observed for mannosidase (MAN) activity. Moreover, notable changes in the activity of β-galactosidase (β-GAL), α-galactosidase (α-GAL), as well as acid phosphatase (ACP) were not statistically confirmed.

The results of antioxidant enzymes in the ruminal fluid are demonstrated in Table 2. The feeding period decreased catalase (CAT) activity (*p* < 0.01) in the experimental group. However, no significant changes in the glutathione reductase (GR) activity were noticed, but it can be identified that it has a strong tendency to increase. Changes in the activity of superoxide dismutase (SOD) were not statistically confirmed. 

The results obtained for non-enzymatic compounds in rumen fluid are shown in Table 3. Significant reduction (*p* < 0.01) from 140.00 to 118.00 was noticed for GSH activity. However, increased concentration was observed for malondialdehyde (MDA) from 0.36 to 0.45 and vitamin C from 22.90 to 24.10 in the experimental group. There were no significant differences in the level of DPPH (2,2-diphenyl-1-picrylhydrazyl) and polyphenols between groups.

The highest correlations were observed between AlaAP and LeuAP (r = 0.897) in the aminopeptidases group and between β-Gal and MAN (r = 0.880) in the glycosidases group (Figure 1.). Furthermore, rumen aminopeptidases were positively correlated with all the investigated glycosidases and superoxide dismutase (SOD) after feeding the cows with dried apple pomace as a feed additive. The most significant correlation were observed for AlaAP and AcP (r = 0.695), LeuAP and MAN (r = 0.588) and ArgAP and β-Gal (r = 0.646). In turn, among glycosidases significant correlations were determined between β-Gal and MDA (r = 0.263), polyphenols (r = 0.237), vitamin C (r = 0.299), and GR (r = 0.200). The most negative dependences were shown between BGRD and DPPH (r = −0.185), GSH (r = −0.185) and catalase (r = −0.298). Significant positive correlations were found between DPPH and HEX (r = 0.439) and polyphenols (r = 0.369), while negative ones for three glycosidases such as α-Gal (r = −0.533), α-Glu (r = −0.246), ACP (r = −0.306), one aminopeptidase such as AlaAP (r = −0.240), and SOD (r = −0.416). The study showed that there are no significant positive correlations between GSH and lysosomal enzymes except between ACP (r = 0.319) and α-Gal (r = 0.254). Positive correlation was also observed with CAT (r = 0.271), while negative between MAN (r = −0.270), polyphenols (r = −0.390), and GR (r = −0.255). Besides, AcP was negatively correlated with MDA (r = −0.274) and polyphenols (r = −0.463). Moreover, the MDA was positively correlated with polyphenols (r = 0.318), vitamin C (r = 0.337), and MAN (r = 0.278). There was also a positive correlation between polyphenols and HEX (r = 0.244), vitamin C (r = 0.274), while negative with α-Gal (r = −0.345), GSH (r = −0.390). Among the enzymatic antioxidants, the most positive correlations were established between catalase and vitamin C (r = 0.288). In contrast, the negative dependence was determined between SOD and polyphenols (r = −0.232) and between CAT and GR (r = −0.224). Vitamin C was also positive correlated with BGRD (r = 0.240), α-Glu (r = 0.259) and MAN (r = 0.258). SOD was significant correlated with ACP (r = 0.450), BGRD (r = 0.300), α-Gal (r = 0.408), and α-Glu (r = 0.351). Significant correlation was also found between CAT and HEX (r = 0.255), and β-Glu (r = 0.368).

## 3. Discussion 

Many studies confirmed that the inclusion of fruit pomace in animal nutrition is necessary and beneficial for the environment as well as for animal health [18,19,20]. Grape pomace extract improves the immune and health status of cows due to the content of polyphenols, mainly resveratrol [21,22]. In turn, the composition of ensiled apple and tomato pomace, which is the source of lycopene [23], leads to a decrease in chewing activity and rumen pH as a consequence of enhancing the production of short-chain fatty acids by microbiota [16,24,25]. However, Steyn et al. [26] showed that dried apple pomace (DAP) had no impact on rumen fermentation. Overall, this study detected differences in the antioxidant capacity in the ruminal fluid between the groups. 

Positive correlations between aminopeptidases, glycosidases, and acid phosphatase (AcP) indicate their interaction in the ruminal fluid lysosomal degradation. Aminopeptidases belong to the lysosomal enzymes that are involved in the maintenance of cellular homeostasis. These exopeptidases are responsible for the accumulation and rotation of proteins as well as micronutrients in an animal body [27]. Furthermore, they are accountable for the degradation of a protein in the final digestion [28]. The fluctuation of aminopeptidase activity can be observed during changes in feeding throughout the addition of some nutrients. The addition of sodium butyrate to the forage decreased the activity of aminopeptidase in the distal jejunum of sheep [29]. The same results were obtained after supplementation with Cu and Zn nanoparticles in turkeys [30], while the addition of organic selenium did not affect the aminopeptidase activity [31]. Furthermore, variable enzyme activity has been noticed depending on the feeding season [32]. Recent studies have shown that increased aminopeptidase activity correlates with renal dysfunction [33] and mammary gland inflammation [34] or can be initial biomarkers of cardiovascular disease in animal models, such as mice and rats [35]. Apple pomace is known as a low protein source, including essential amino acids [7]. It has been shown that in comparison to ensiled apple pomace, lower protein content has dried apple pomace [36], which may explain the results obtained in the experimental group. Decreased enzymatic activity may indicate a reduction in protein degradation or protein digestibility. It can be assumed that the addition of dried apple pomace leads to disturbances in protein turnover. On the other hand, the processes of rumen protein degradation were not so notable due to activity changes in one enzyme belonging to the aminopeptidases. However, the ruminants can recycle urea-N to the rumen by increasing the nitrogen efficiency for a low protein diet. Another reason for maintaining nitrogen efficiency is the reduction of the catabolism of protein for energy [37]. It should be mentioned that protozoa are also involved in protein metabolism [38]. The high content of carbohydrates, including fiber in apple pomace, may impact microorganisms inhabiting rumen and can be perceived as eubiotic properties for the rumen.

The role of glycosidases is to hydrolysate glycosidic linkages in oligo- or polysaccharides, glycolipids, glycoproteins, and glycoconjugates. These catalysts play a pivotal role in biological processes, but all functions are still not completely recognized [39]. Replacement of the control diet with a diet enriched in dried apple pomace caused changes in glucosidase activity in the rumen of the experimental group. Apple pomace is a rich source of carbohydrates and contains a higher amount of this nutrient compared to apples [40]. A significantly lower activity of HEX and β-Glu in the experimental group can be associated with the high amount of reducing sugars/simple carbohydrates like glucose and fructose [41]. The role of HEX is to degrade glycoconjugates by cleaving N-acetyl-D-glucosamine and N-acetyl-D-galactosamine from the non-reducing ends of oligosaccharide chains of glycoproteins, glycolipids as well as glycosaminoglycans [42]. β-Glu is also involved in the degradation of non-reducing terminal glucosyl residues [43]. In turn, a higher activity of glycosidases (BGRD, MAN, and α-Glu) was probably linked with a more effective degradation process in the case of oligosaccharides (disaccharides and glucuronides). Interesting results were determined for the correlations between some glycosidases (HEX, β-Gal, and β-Glu) and the polyphenols. The study by Ajila et al. [44] demonstrated that β-glucosidases play a crucial role in the release of polyphenolic aglycones from apple pomace. Therefore, the positive correlations in this study indicate that these enzymes may increase the antioxidant capacity of apple pomace. 

In the present study, changes in reduced glutathione GSH levels were also observed. Glutathione belongs to the endogenous antioxidants that are involved in the regulation of cellular metabolism. The protective impact is associated with the ability to detoxification of free radicals, hydrogen/organic peroxides, and other harmful reactive forms of oxygen. Glutathione occurs, among others, in the form of reduced (GSH) and oxidized (GSSG) glutathione. Reduced form (GSH) plays a crucial role in enzymatic reactions as a cofactor, and it is very significant for the regeneration of other antioxidants, including tocopherols or ascorbate [45,46,47]. The decrease in reduced glutathione (GSH) level in the experimental group is linked with glutathione reductase (GR) and vitamin C. The main role of GR is to regulate, modulate, and maintain cellular redox homeostasis. The positive correlation between GR and GSH is explained by studies in which one of the roles of GR is to maintain the supply of GSH [48]. The results indicated that dried apple pomace as an additive to ruminant feedstuff exerts the stimulation of oxidation–reduction potential reduction potential due to increasing concentration of glutathione reductase (GR) as well as vitamin C. The concentration of DDPH in the rumen is not changeable during the feeding; thus, the vitamin C and glutathione reductase are sufficient factors to maintain the redox process. Furthermore, GSH and vitamin C show a synergistic effect because of common targets in their functions against oxidative damage and the vitamins can compensate for GSH depletion [49]. Therefore, vitamin C contained in dried apple pomace as a low-molecular-weight compound is sufficient to protect against oxidative processes in the rumen, and the first line (enzymatic) of defense is not activated. It is confirmed by a lower level of catalase activity, which is involved in detoxifying secondary free radicals [50]. On the other hand, a significant correlation indicates that catalase activity is stimulated by polyphenols. Some studies have proven that dried apple pomace is a rich source of polyphenols [51,52]. The main compounds include flavan-3-ols, (−)-epicatechin as well as (+)-catechin, and their polymers are referred to as procyanidins, flavonols (quercetin glycosylated derivatives), dihydrochalcones, and phenolic acids (e.g., caffeic acid) [53,54,55]. This may suggest that apple polyphenols are beneficial for the health of farm animals. Xu et al. [56] suggested that lipid metabolism is improved in weaned piglets due to these antioxidants. Other studies have shown that polyphenolic compounds can significantly modify rumen fermentation by modulating the microbial population, mainly the protozoa community [57,58]. The inclusion of fruit pomace with polyphenols in feedstuff affects not only the antioxidant system in rumen fluid. Besides, it has more significant benefits. Studies have shown that polyphenols influence animal reproduction processes due to preventing reproductive cells from oxidation damage. Even low concentrations of polyphenols may neutralize free radicals improving embryonic development. Nevertheless, isoflavones can lead to hormonal imbalances, reducing farm animals’ fertility [59]. Furthermore, polyphenols are metabolized by the microbial community to volatile fatty acids (VFA), such as propionic acid, acetic acid, and butyric acid, which are the main source of energy for ruminants. The adequate amount of energy from forage with polyphenols addition can reveal cow genetic potential and modulate milk yield and components in milk (e.g., protein, lactose) [60]. Polyphenols can replace the role of enzyme compounds, as confirmed by a negative correlation with SOD. The function of superoxide dismutase (SOD) is to degrade superoxide O_2_^−^ into oxygen and hydrogen peroxide; therefore, it is known as an endogenous cellular defense system [61]. Thus, vitamin C and polyphenols can inhibit the enzymatic antioxidant system in the ruminal fluid. However, a high correlation between SOD and DPPH shows changes in the oxidation–reduction potential of the rumen. A higher concentration of MDA can indicate an increased lipid oxidative mechanism in the rumen. MDA is one of the best-known marker of polyunsaturated fatty acids peroxidation in the cells [62]. This may be related to the digestion or degradation of the lipid membrane of the rumen microflora and glycoproteins due to an increase in the levels of α-glucosidase (α-Glu) and mannosidase (MAN). It can be linked to the protozoa community and their role in microbial lipolytic activity [31]. On the other hand, some studies showed that cows fed a diet containing fermentable carbohydrates were more exposed to an increase in MDA levels in the ruminal fluids [63,64]. The apple pomace is a valuable fermentation substrate due to its total sugar content, especially fructose. Therefore, dried apple pomace as a feedstuff additive can intensify fermentation processes in the rumen. It has to be also mentioned that not only diet affects the redox state. The oxidation–reduction potential is related to microbial fermentation activity and includes quantitative and qualitative changes in the microorganisms, microbial growth rate, and fermentation processes. The reduction of potential redox influences the intensity of fermentation pathways and, thus, the growth of the microbial community in the rumen [65]. 

## 4. Materials and Methods 

### 4.1. Animals

The study was conducted during the spring season (April–May 2021) at the dairy cow farm in Szemborowo, Poland. The experiment included 4 Polish Holstein–Friesian cannulated dairy cows of about 625 kg (+/−20 kg) body weight, 4–5 months in lactation with an average of 33 ± 2.1 kg/d milk production. The animals were housed in separated tie-stalls. The cows were free from any disease symptoms.

### 4.2. Experimental Plan/Design 

The study was planned in accordance with the replicated 2 × 2 crossover design and lasted 52 days, including two cycles (2 × 26 days). Each of them consisted of a 21-day adaptation period and a 5-day sample collection period for further analysis. A total length of the experiment is an acceptable period for the rumen microbiota population to become stable in experimental conditions (e.g., supplementation of experimental factors such as a source of bioactive component). After this time, rumen metabolism is stabilized and sample collection can be done. Sometimes even a shorter period has been applied, such as a 14 or 15-day adaptation period [66,67,68]. The control group was fed a standard diet (TMR-total mixed ration), while the experimental group received TMR + 6% (150 g/kg of DM) of dried apple pomace as a feed additive. Access to fodder (at 6 a.m. and 6 p.m.) and water was available ad libitum. The feedstuff and dried apple pomace composition are presented in Table 4 and Table 5. The feeding protocol used in our experiment is the preferred approach used in this type of experiment based on ruminants. Such protocol is not only used in nutritional studies but also when, e.g., microbial analysis is done [69,70,71].

### 4.3. Collecting of Rumen Fluid

The ruminal fluid samples (about 400 g per cow) were taken from the midventral sac in three locations: top, bottom, and middle. Subsequently, the samples were clarified by using two layers of cheesecloth. Afterward, rumen fluids were frozen and stored at −20 °C for further chemical analysis. Prior to performing the final analyses, rumen samples collected from the same cow on the same day at 0 h, 3 h, and 6 h post-feeding were thawed and mixed to obtain a more representative sample of the rumen fluid [72].

### 4.4. Measurement of Lysosomal Enzymes Activity 

Rumen fluid samples were thawed/frozen three times before measuring the activity of lysosomal enzymes. Subsequently, to obtain the supernatant, samples were separated by centrifugation at 4 °C (1500× *g*, 15 min). The activities of alanyl-aminopeptidase (AlaAP-EC 3.4.11.2), leucyl-aminopeptidase (LeuAP-EC 3.4.11.1), and arginyl-aminopeptidase (ArgAP-EC 3.4.11.6) were established at 540 nm after incubation at 37 °C for 1 h, according to the method of McDonald et al. [73]. The activities of acid phosphatase (AcP–EC 3.1.3.2), β-glucuronidase (BGRD-EC 3.2.1.31), α-glucosidase (α-Glu EC 3.2.1.20), β-glucosidase (β-Glu-EC 3.2.1.21), β-galactosidase (β-Gal-EC 3.2.1.23), N-acetyl-hexosaminidase (HEX), α-galactosidase (α-Gal), and mannosidase (MAN-EC 3.2.1.24) were measured as 4-nitrophenyl derivatives after incubation at 37 °C at 420 nm in accord with Barret and Heath [74] method. The spectrophotometer UV-VIS CarryBio 50 CarryBio 50 (Santa Clara, CA, USA) was used for the measurement. The enzyme activity was presented in nmol/mg protein/h.

### 4.5. Superoxide Dismutase (SOD E.C 1.15.1.1) Activity 

Rumen fluid samples were centrifuged at 1000× *g* for 10 min at 4 °C. Afterwards, the samples were homogenized in 5 mL cold 20 mM HEPES buffer at pH 7.2 (containing 1 nM EGTA, 210 mM mannitol, and 70 MN sucrose). The homogenates were centrifuged at 1500× *g* for 5 min at 4 °C. Samples were kept on ice until the beginning of analysis to avoid uncontrolled reaction initiation. The SOD analysis was conducted using the Superoxide Dismutase Assay Kit, Item No. 706002 (Cayman Chemical Company; Ann Arbor, MI, USA). The absorbance was measured at 460 nm using a microplate reader Synergy4 (BioTek, Winooski, VT, USA). Calculations of SOD activity were conducted with Gen5 software created by Biotek. The activity of the enzyme was presented in U/mL. 

### 4.6. Glutathione Reductase (GR E.C 1.8.1.7) Activity 

Rumen supernatant after centrifugation at 2000× *g* for 10 min at 4 °C was homogenized in cold potassium phosphate buffer, pH 7.5. Subsequently, the samples were centrifugated at 10,000× *g* for 15 min at 4 °C. The supernatant was stored on ice until the beginning of the analysis. Further steps were followed according to the guidance of the producer of Assay Kit Item No. 703202 (Cayman Chemical Company; Ann Arbor, Michigan, USA). The absorbance was measured at 340 nm. A standard curve was formed using pure glutathione reductase (Sigma G4751). To measure the GR activity, the microplate reader Synergy4 by BioTek was used. The results were determined using the Gen5 software. The activity of the enzyme was expressed in nmol/min/mL.

### 4.7. Catalase (CAT E.C 1.11.1.6) Activity

Rumen samples were centrifuged at 2000× *g* for 10 min at 4 °C and then homogenized in cold 50 mM potassium phosphatase buffer containing 1 mM EDTA. Subsequently, the samples were centrifuged at 10,000× *g* for 15 min at 4 °C. Obtained supernatants were stored on ice until the beginning of the analysis. The next stages were performed according to the instructions of the kit’s producer, Item No.707002 (Cayman Chemical Company). The measurement of absorbance at 540 nm was carried out with a microplate reader Synergy4 (BioTek). A standard curve was created using pure formaldehyde provided by the producer. The results were established using the Gen5 software. The CAT activity was presented in nmol/min/mL. 

### 4.8. Measurement of Malondialdehyde (MDA) Level

Before starting the procedure, the rumen fluid was thawed/frozen three times. Subsequently, samples were homogenized in a phosphate buffer and centrifuged at 3000× *g* for 10 min at 4 °C. The supernatant was kept on ice until the start of the analysis. The further procedure was conducted according to the guidance of the producer OxisReasearch™ Bioxytech^®^ MDA-586 ™ test (Foster City, California 94404-1136 USA). The absorbance was measured at 586 nm using the Cary Varian 50Bio spectrometer (Santa Clara, CA 95051 USA). Calculations were presented based on a calibration curve received according to the producer’s recommendations and the template included in the test report. The level of MDA was presented in µM.

### 4.9. Reduced Glutathione (GSH) Content 

The concentration of GSH was conducted using the OxisResearch™ Bioxytech^®^ GSH/GSSG—412™ test (Foster City, CA, USA). Preparation of samples for further analysis required refrigeration with the addition of M2VP (1-methyl-2-vinyl-pyridium trifluoromethanesulfonate) at −80 °C. The procedure was performed according to instructions delivered by the kit’s producer. The measurement of absorbance was carried out using the microplate reader Synergy4 at 412 nm. The results were determined using the Gen5 software. The concentration of GSH was expressed in thiol groups (mmol-SH groups). 

### 4.10. Level of Vitamin C (Ascorbic Acid) 

Rumen fluid was homogenized in phosphate buffer at pH 7. Afterward, the 10% trichloric acid was added and the samples were centrifuged at 3000× *g* for 10 min. The obtained supernatant was mixed with phosphoric acid V, 2,2′-bipyridyl, and iron chloride (III) and incubated for one hour at 37 °C. Absorbance was measured at 525 nm. Solutions of vitamin C standard (O.5–5 mg of vitamin C A92902 Sigma-Aldrich L-Ascorbic acid 99%) were performed similarly. The spectrophotometer UV-VIS CarryBio 50 (Santa Clara, CA, USA) was used for the measurement. The results were expressed in mg/100 mL.

### 4.11. Total Polyphenols Content 

Rumen fluid was homogenized in ultra-pure methanol with 1% acetic acid. Then, the samples were extracted for 2 h in an ultrasonic bath at 40 °C and then centrifuged at 4000× *g* for 10 min at 4 °C. Total polyphenols content was performed according to the modified protocol of Škerget et al. [75], which is based on the colorimetric oxidation–reduction potential reduction reaction. The supernatant was transferred to a 6-mL test tube and mixed with 2.5 mL of the Folin–Ciocalteu reagent (AOCS, 1990), which was 10-fold diluted with demineralized water (Sigma-Aldrich, Buchs, Switzerland). Prior to the incubation at 40 °C for 30 min, the sodium carbonate solution was added. Subsequently, the absorbance of samples was measured at 765 and 735 nm. In this procedure, the blank sample consisted of 0.5 mL of ddH_2_O instead ruminal fluid. Calculations were conducted using a calibration curve prepared based on the absorbance of the gallic acid standard in the range of 0 to 0.5 mg/mL. The results were presented in mg of GAE/mL ruminal fluid (GAE–gallic acid equivalent). 

### 4.12. Potential to Scavenge the Free DPPH Radical 

The antioxidative activity of the analyzed material was carried out according to a modified procedure of Brand-Williams et al. [76], using a synthetic DPPH radical (1,1-diphenyl-2-picrylhydrazyl). Samples were homogenized in ultra-pure methanol with 1% acetic acid. Then, the homogenates were extracted for two hours in an ultrasonic bath at 40 °C. Afterward, the tubes were centrifuged at 4000× *g* for 10 min at 4 °C. 0.5 mL of each obtained supernatant was collected and mixed with 0.5 mL of an ethanolic solution of 1,1-diphenyl-2-picrylhydrazyl (0.5 mM). Before adding, it was diluted to confirm its absorbance of ca. 0.9 at 517 nm. The samples were kept in a dark and cool place for 30 min to stabilize the color. The spectrophotometer UV-VIS CarryBio 50 (Santa Clara, CA, USA) was used to measure the wavelength at 517 nm.

### 4.13. Statistical Analysis

The normality of data test was performed using the PROC UNIVARIATE procedure in SAS. Based on results from the Shapiro–Wilk test, an independent t-test (parametric test; PROC TTEST procedure) or independent-samples Mann–Whitney U test (non-parametric test; PROC NPAR1WAY procedure), depending on the parameters, was performed using SAS. The results were considered significant when the *p*-values were less than 0.05 or 0.01. All values are presented as means with pooled standard errors of the means.

## 5. Conclusions 

Dried apple pomace is a beneficial feed additive in cows’ diets due to its valuable nutrients and antioxidant compounds, mainly polyphenols. The inclusion of fruit pomace in daily feeding enhances oxidation–reduction potential and modulates lysosomal degradation processes in the ruminal fluid. Low-weight compounds (e.g., vitamin C) in dried apple pomace are sufficient factors to protect rumen against the oxidation process, as shown by our results, but it is important to confirm these results under conditions where not only rumen will be evaluated but also products like milk.

## Figures and Tables

**Figure 1 ijms-23-10475-f001:**
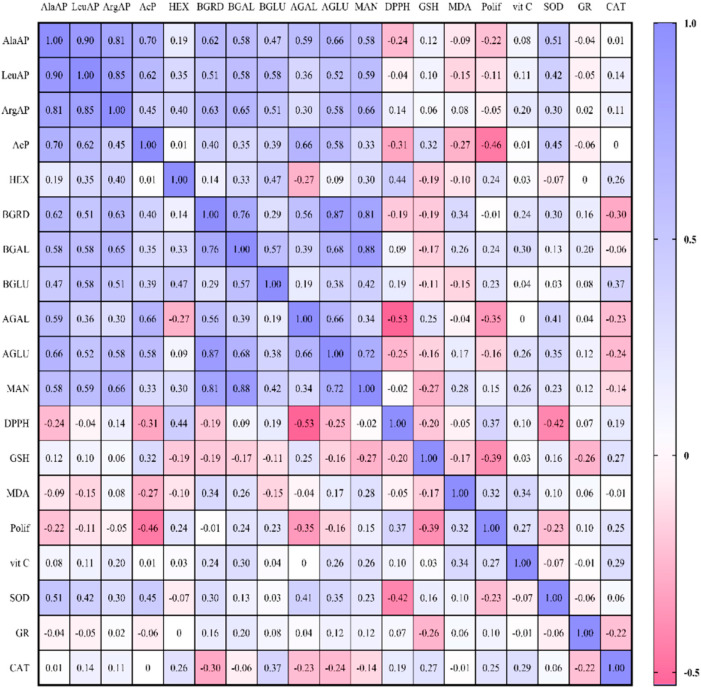
Significant Pearson correlation (*p* < 0.01) and (*p* < 0.05) between feeding system.

**Table 1 ijms-23-10475-t001:** The activity (nmol/mg protein/h) of lysosomal enzymes in ruminal fluid in the control and experimental group.

	CON	EXP	SEM	*p*-Value *
AlaAP	154.90	140.90	3.61	0.10
LeuAP	163.70	143.10	3.48	<0.01 **
ArgAP	157.60	159.10	2.91	0.15
AcP	1618.00	1412.00	41.50	0.11
HEX	231.00	194.00	6.11	<0.01 **
BGRD	31.10	42.40	1.48	<0.01 **
β-GAL	163.00	186.00	6.31	0.07
β-GLU	1294.00	1136.00	25.60	<0.01 **
α-GAL	368.00	357.00	17.20	0.89
α-GLU	245.00	327.00	13.20	<0.01 **
MAN	29.70	36.70	1.17	<0.01 **

** Correlation is significant at the 0.01 level (2-tailed). * Correlation is significant at the 0.05 level (2-tailed). CON—control group. EXP—experimental group.

**Table 2 ijms-23-10475-t002:** The activity of antioxidant enzymes in the ruminal fluid in the control and experimental group.

	Unit	CON	EXP	SEM	*p*-Value *
SOD	U/mL	2.19	2.15	0.06	0.52
GR	nmol/min/mL	16.10	20.80	1.19	0.06
CAT	nmol/min/mL	93.20	68.80	3.72	<0.01 **

** Correlation is significant at the 0.01 level (2-tailed). * Correlation is significant at the 0.05 level (2-tailed). CON—control group. EXP—experimental group.

**Table 3 ijms-23-10475-t003:** The activity of non-enzymatic compounds in ruminal fluid in control and experimental group.

	Unit	CON	EXP	SEM	*p*-Value *
DPPH	% of remaining DPPH	73.20	72.80	0.39	0.95
GSH	µM	140.00	118.00	2.65	<0.01 **
MDA	µM	0.36	0.45	0.01	<0.01 **
Polyphenols	mg/GAE/mL	1.38	1.40	0.01	0.16
Vit C	mg/100	22.90	24.10	0.25	<0.01 **

** Correlation is significant at the 0.01 level (two-tailed). * Correlation is significant at the 0.05 level (2-tailed). CON—Control Group; EXP—Experimental Group.

**Table 4 ijms-23-10475-t004:** The composition of the standard and experimental diets (with the additive of dried apple pomace) was dedicated to cows during the experiment.

Items	Unit	Control Diet	Experimental Diet
Dry matter (DM), kg/kg as fed	Kg	23.80	25.40
Metabolizable Energy (ME)	MJ	255.00	245.00
Net Energy Lactation (NEL)	MJ	157.00	151.00
Crude protein (CP), g/kg DM	G	3505.00	3515.00
Utilizable protein, g/kg DM	G	3310.00	3174.00
Total undegradable protein in rumen, g/kg DM	G	794.00	661.00
Total fiber, g/kg DM	G	3285.00	4013.00
Neutral detergent fiber (aNDF), g/kg DM	%	14.09	13.52
Digestibility of the dry matter dose (IVDMD, %)	%	78.08	78.82
Percentage of dry matter intake/cow weight	%	4.33	4.65
Relative Feedstuff Value (RFV) index		262.00	284.00
Starch, g/kg DM	G	7963	7211
Calcium, g/kg of DM	G	69.90	67.20
Phosphorus, g/kg of DM	G	72.60	67.70
Calcium: Phosphorus		01:01	01:01
Magnesium, g/kg of DM	G	38.40	35.00
Sodium, g/kg of DM	G	15.00	13.60
Zinc, g/kg of DM	Mg	189.00	157.00
Copper, g/kg of DM	Mg	39.90	33.20
Manganese, g/kg of DM	Mg	157.00	131.00
Cobalt, g/kg of DM	Mg	0.63	0.49
Iodine, g/kg of DM	Mg	105.00	87.50
Selenium, g/kg of DM	Mg	0.28	0.21
Vitamin A, IU/kg	j.m.	18,900.00	15,750.00
Vitamin D, IU/kg	j.m.	5906.00	4922.00
Vitamin E, IU/kg	Mg	77.70	64.70
Niacin, g/kg of DM	Mg	126.00	105.00
Biotin, g/kg of DM	µg	550.00	458.00
Total polyphenols	mgGEA/g	6.67	6.85

**Table 5 ijms-23-10475-t005:** Nutritional value of the dried apple pomace (DAP) used in the experiment.

Dried Apple Pomace Composition		Values	SEM
Metabolic Energy	MJ/kg	11.50	1.01
Protein	%	11.60	1.21
Fat	%	5.10	0.48
Fiber	%	60.00	5.56
Total Sugars	%	16.90	2.02
Ash	%	2.20	0.18
Moisture content	%	4.30	0.37

## Data Availability

The data presented in this study is contained within the article.

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
