# Peer review of "Effect of Dried Apple Pomace (DAP) as a Feed Additive on Antioxidant System in the Rumen Fluid"

_ijms, 2022, doi:10.3390/ijms231810475_

Round 1
Reviewer 1 Report
In this manuscript, the authors reported that the addition of dried apple pomace (DAP) can be used to maintain redox balance in the rumen. Through 52 days of feeding, they found that the activity of lysosomal enzymes decreased, while the activity of antioxidant enzymes increased in rumen fluid. They therefore concluded that apple pomace as a feed additive may have potential value in practical application in animal husbandry. Although the results appear to be quite positive, since the primary purpose of this study should be to demonstrate the benefits of DAP on the gastrointestinal tract, in all its presented data, there is absolutely no specification for the animals tested, including tissue oxidative stress or inflammation changes. This makes the benefits of DAP less compelling. In addition, lack of experimental design or data on molecular mechanisms is also a major disadvantage, and the description of some experimental steps is also not very clear. These comments are listed below for reference.
1. Experimental design: The authors stated that the feeding protocols were for 52 days (2x26 days), and 6% DAP was used as the feed for the experimental group. What is the basis for setting such a time schedule and food formula? Are there any pre-experiments or previous literature to support this?
2. Tab. 1: Since the authors deduced that polyphenols and other low-weight antioxidant compounds may be the main reason for their effectiveness, it is suggested that the levels of these substances should be clearly listed in this table.
3. Collecting of rumen fluid: The authors claim that the time points for collecting rumen fluid are before 0h, 3h, and 6h post-feeding, but the follow-up data does not indicate which time point, or the comparative difference between the time points. Please clarify.
4. Fig. 1: The significance of this result is unclear. Although the activity of lysosomal enzymes is roughly dependent (blue square area), this trend is lacking in activity of non-enzymatic compounds. In other words, the results do not seem to be completely consistent with the authors' inferences, and it is necessary to clarify.
5. In my opinion, the redox state of the gastrointestinal tract is not only related to food, but also the intestinal flora ecology seems to have a great relationship. However, in all of the authors' texts this doesn't seem to take this into account.
6. The content of Author contributions is manifestly incorrect.
Author Response
We thank you very much for the careful revision to our manuscript. We realized from the comments received that several key points in our original work have not been properly addressed. Additionally, we also recognized and appreciated the competence of the Reviewers. We have incorporated most of the suggestions of the reviewers that certainly improved the quality of our manuscript
In this resubmitted version, we took advantage of the criticisms received and modified the text accordingly, adding explanations and new experimental results.
Please find our point-by-point responses color-coded in red.

Reviewer 2 Report
Dried apple pomace is a beneficial feed additive in cows' diet due to its valuable nutrients and antioxidant compounds, mainly polyphenols.
Inclusion of fruit pomace in daily feeding enhances oxidation-reduction potential and modulates lysosomal degradation processes in the ruminal fluid.
Low-weight compounds (e.g. vitamin C) in dried apple pomace are sufficient factors to protect rumen against the oxidation process, as shown by our results,
It is important to confirm these results under conditions where not only rumen will be evaluated but also products like milk.

Author Response
We thank you very much for the careful revision to our manuscript.
Kind regards,
Magdalena Koszarska

Round 2
Reviewer 1 Report
Thanks to the authors for their careful replies, and the manuscript has been significantly improved in this revised version, but I still have two minor questions I would like to ask the authors,
1) The authors cited many citations in the response as evidence, but the vast majority of these articles seem to have not been included in this revised version. I don't think the author's replies can only be aimed at reviewers. If the content is reasonable, they should be considered to be appropriately included in the revised version.
2) As with my last question, the authors don't seem to have responded. Even though they were demonstrated giving DAP may have improved the antioxidant system in the rumen fluid. But from an overall animal point of view, did these weeks of feeding make the animals healthier or gain other significant benefits? I think it is necessary for the authors to add some more arguments to this point to fit the topic.
